# Learning to Solve Multi-Robot Task Allocation with a Covariant-Attention based Neural Architecture

## Abstract

This paper presents a new graph neural network architecture over which reinforcement learning can be performed to yield online policies for an important class of multi-robot task allocation (MRTA) problems, one that involves tasks with deadlines, and robots with ferry range and payload constraints and multi-tour capability. While drawing motivation from recent graph learning methods that learn to solve combinatorial optimization problems of the mTSP/VRP type, this paper seeks to provide better convergence and generalizability specifically for MRTA problems. The proposed neural architecture, called Covariant Attention-based Model or CAM, includes three main components: 1) an encoder: a covariant compositional node-based embedding is used to represent each task as a learnable feature vector in manner that preserves the local structure of the task graph while being invariant to the ordering of graph nodes; 2) context: a vector representation of the mission time and state of the concerned robot and its peers; and 2) a decoder: builds upon the attention mechanism to facilitate a sequential output. In order to train the CAM model, a policy-gradient method based on REINFORCE is used. While the new architecture can solve the broad class of MRTA problems stated above, to demonstrate real-world applicability we use a multi-unmanned aerial vehicle or multi-UAV-based flood response problem for evaluation purposes. For comparison, the well-known attention-based approach (designed to solve mTSP/VRP problems) is extended and applied to the MRTA problem, as a baseline. The results show that the proposed CAM method is not only superior to the baseline AM method in terms of the cost function (over training and unseen test scenarios), but also provide significantly faster convergence and yields learnt policies that can be executed within 2.4ms/robot, thereby allowing real-time application.

## 1 Introduction

In multi-robot task allocation (MRTA) problems, we study how to coordinate tasks among a team of cooperative robotic systems such that the decisions are free of conflict and optimize a quantity of interest (Gerkey & Matarić, 2004). The potential real-world applications of MRTA are immense, considering that multi-robotics is one of the most important emerging directions of robotics research and development (Yang et al., 2018; Rizk et al., 2019), and task allocation is fundamental to most multi-robotic or swarm-robotic operations. Example applications include disaster response (Ghassemi & Chowdhury, 2018), last-mile delivery (Aurambout et al., 2019), environment monitoring (Espina et al., 2011), and reconnaissance (Olson et al., 2012)). Although various approaches (e.g., graph-based methods (Ghassemi & Chowdhury, 2018; Ghassemi et al., 2019), integer-linear programming (ILP) approaches (Nallusamy et al., 2009; Toth & Vigo, 2014; Cattaruzza et al., 2016; Jose & Pratihar, 2016), and auction-based methods (Dias et al., 2006; Schneider et al., 2015)) have been proposed to solve the ***combinatorial optimization*** problem underlying MRTA operations, they usually do not scale well with number of robots and/or tasks, and do not readily adapt to complex problem characteristics without tedious hand-crafting of the underlying heuristics. In the recent years, a rich body of work has emerged on using learning-based techniques to model solutions or intelligent heuristics for combinatorial optimization (CO) problems over graphs. The existing methods are mostly limited to classical CO problems, such as multi-traveling salesman (mTSP), vehicle

routing (VRP), and max-cut type of problems. In this paper, we are instead interested in learning policies for an important class of MRTA problems (Korsah et al., 2013) that include characteristics such as tasks with time deadlines, robots with constrained payload and ferry-range, and ability to conduct multiple-tours.

In this paper, we show how such MRTA problems can be modeled as a Markov Decision Process over graphs, allowing us to learn task allocation policies by performing reinforcement learning (RL) over graphs. We specifically focus on a class of MRTA problems that falls into the Single-task Robots, and Single-robot Tasks (SR-ST) class defined in (Gerkey & Matarić, 2004; Nunes et al., 2017). Here, a feasible and conflict-free task allocation is defined as assigning any task to only one robot (Ghassemi et al., 2019)). Subsequently, we propose a new **covariant attention-based model** (aka **CAM**), a neural architecture for learning over graphs to construct the MRTA policies. This architecture builds upon the attention mechanism concept and innovatively integrates an equivariant embedding of the graph to capture graph structure while remaining agnostic to node ordering.

## 1.1 MULTI-ROBOT TASK ALLOCATION

The MRTA problem can be formulated as an Integer Linear Programming (ILP) or mixed ILP. When tasks are defined in terms of location, the MRTA problem becomes analogical to the Multi-Traveling Salesmen Problem (mTSP) (Khamis et al., 2015) and its generalized version, the Vehicle Route Planning (VRP) problem (Dantzig & Ramser, 1959). Existing solutions to mTSP and VRP problems in the literature (Bektas, 2006; Braekers et al., 2016) have addressed analogical problem characteristics of interest to MRTA, albeit in a disparate manner; these characteristics include limited vehicle capacity, tasks with time deadlines, and multiple tours per vehicle, with applications in the operations research and logistics communities (Azi et al., 2010; Wang et al., 2018). ILP-based mTSP-type formulations and solution methods have also been extended to task allocation problems in the multi-robotic domain (Jose & Pratihar, 2016). Although the ILP-based approaches can in theory provide optimal solutions, they are characterized by exploding computational effort as the number of robots and tasks increases (Toth & Vigo, 2014; Cattaruzza et al., 2016). For example, for the studied SR-ST problem, the cost of solving the exact ILP formulation of the problem, even with a linear cost function (thus an ILP), scales with $\mathcal{O}(n^3 m^2 h^2)$, where $n$, $m$, and $h$ represent the number of tasks, the number of robots, and the maximum number of tours per robot, respectively (Ghassemi et al., 2019). As a result, most practical online MRTA methods, e.g., auction-based methods (Dias et al., 2006) and bi-graph matching methods (Ghassemi & Chowdhury, 2018), use some sort of heuristics, and often report the optimality gap at least for smaller test cases compared to the exact ILP solutions. Recently, it has been shown that Graph Neural Networks (GNNs) can provide an alternative method with a computational efficient run-time (Kool et al., 2019).

## 1.2 LEARNING OVER GRAPHS

Neural network based methods for learning CO can be broadly classified into: (i) Reinforcement Learning (RL) methods (Kool et al., 2019; Barrett et al., 2019; Khalil et al., 2017); and (ii) supervised learning (often combined with RL) methods (Kaempfer & Wolf, 2018; Mittal et al., 2019; Li et al., 2018; Nowak et al., 2017). The supervised learning approaches typically address problem scenarios where samples are abundant (e.g., influence maximization in social networks (Mittal et al., 2019)) or inexpensive to evaluate (e.g., TSP (Kaempfer & Wolf, 2018)), and are thus unlikely to be readily applicable to solve complex problems over real-world graphs. RL based techniques to learn on graphs include attention models with REINFORCE (Kool et al., 2019) and deep Q-learning (Khalil et al., 2017; Barrett et al., 2019), among others, with some extending solutions to multi-agent settings (Jiang et al., 2020) In this work, we are interested in the first class of the methods (i.e., RL methods over graph space).

Dai et al. (2017) showed that a combination of graph embedding and RL methods can be used to approximate optimal solutions for combinatorial optimization problems, as long as the training and test samples are drawn from the same distribution. Mittal et al. (2019) presented a new framework to solve a combinatorial optimization problem. In this framework, Graph Convolutional Network (GCN) performs the graph embedding and Q-Learning learns the policy. The results demonstrated that the proposed framework is able to learn to solve unseen test problems that have been drawn from the same distribution as that of the training data-set. More importantly, it has been shown that using a learnt network policy instead of tree search, both methods are using the same embedding GCN, showed a speedup of 5.5 for a problem size of 20,000. Similarly, the effectiveness of learning

a network policy using Q-Learning to solve the Max Cut problem (a combinatorial problem) has been demonstrated by Barrett et al. (2019).

Recently, there has been a growing interest in using sequence-to-sequence models (e.g., pointer networks and attention mechanism) to encode and learn the combinatorial optimization problems in graph space (Kaempfer & Wolf, 2018; Kool et al., 2019). Kool et al. (2019) implemented a framework using an encoder/decoder based on attention mechanism and REINFORCE algorithm for solving a wide variety of combinatorial optimization problem as graphs, with the main contribution being flexibility of the approach on multiple problems with the same hyper parameters. Majority of the above methods solve classical problems (i.e., mTSP, VRP and Max-Cut type of problems) without physical, real-world constraints. Taking into account additional physical characteristics such as robots with payload capacity and range constraints and tasks with time deadline requires fundamental modification of the encoding and decoding mechanisms in the neural architecture, which motivates the research presented here.

In this paper, a new neural architecture is proposed that combines the *attention mechanism* with an enhanced encoding network (embedding layers), where the latter is particularly designed to capture local structural features of the graph in an equivariant manner. The embedding layer is a variation of *Covariant Compositional Networks* (CCN), introduced by Hy et al. (2018). This node-based embedding has been chosen since it: i) operates on undirected graph; ii) uses receptive field and aggregation functions based on tensor product and contraction operations (which are equivariant), which leads to a permutation- and rotation-invariant embedding; and iii) provides an extendable representation ($n$-th order tensor representation can be useful to extend the work to multi-level networks, e.g., involving multiple node properties). We found an exact implementation of the CCN to be computationally burdensome for learning policies in large MRTA problems, and hence a variation of the CCN is proposed in this work.

### 1.3 CONTRIBUTIONS OF THIS PAPER

The general SR-ST problem studied here considers location-encoded tasks, meaning each task is defined as a robot visiting a location and dropping a payload (the payload dropping time is ignored for simplicity of implementation). This setting can also be generalized to multi-agent problems addressing last-mile delivery applications. Unlike a single agent setting, the multi-agent setting leads to a combinatorial optimization problem with additional complexities, attributed to resolving conflicts between agents' decisions. Apart from the multi-agent setting, we also consider additional constraints for tasks and robots/agents. Each task-$i$ in $N$ is associated with a time deadline $t_i$, by which the task has to be done in order for it to be considered as successfully completed. The constraints for the robots include: 1) Each robot has a limited range of distance that it can travel with full charge, i.e., in one tour; and 2) Each robot has a maximum capacity of payload it can carry (bounding the max number of tasks it can perform in one tour).

With the above context, the primary contributions of this paper can be stated as: **1)** Formulating the studied class of MRTA problems as a Markov Decision Process or MDP over graphs, such that the optimal (task allocation) policy can be learned using an RL approach. **2)** Extending the attention-based mechanism (AM) (Kool et al., 2019), serving as a decoder, and associated context to a multi-agent combinatorial optimization setting; and **3)** Incorporating a covariant compositional mechanism for node-based embedding (aka encoder) of the graph such that local structural information and associated task properties are preserved. The resulting CAM architecture is evaluated on a large representative (simulated) MRTA problem, which involves coordinating a team of multiple unmanned aerial vehicles (UAVs) to respond to flood victims by dropping survival packages. The simulation results provide important evidence of the promising convergence of the CAM method, and its advantage over the standard attention mechanism approach, in terms of the MRTA cost function and scalability of the learnt policy with number of tasks. The remaining portion of the paper is organized as follows. Section 2 describes the MRTA problem definition and its formulation as a MDP over graphs. Section 3 presents our proposed new graph neural network for the MRTA problem. Section 5 describes simulation settings and different case studies. Results, encapsulating the performance of these methods on different-sized problems and various analyses of the proposed method, are presented in Section 6. The paper ends with concluding remarks.

## 2 MRTA: PROBLEM DEFINITION AND FORMULATIONS

The multi-robot task allocation (MRTA) problem is defined as the allocation of tasks and resources among several robots that act together without conflict in the same environment to accomplish a

common mission. The optimum solution (decision) of the MRTA problem is a sequence of tasks (conflict-free allocation) that maximizes the mission outcome (or minimize the mission cost) subject to the robots' capacity/trip bounds. Here, the following assumptions are made: 1) All robots are identical and start/end at the same depot; 2) There are no environmental uncertainties; 3) The location $(x_i, y_i)$ of task-$i$ and its time deadline $t_i$ are known to all robots; 4) Each robot can share its state and its world view with other robots; and 5) There is a depot (Task-0), where each robot starts from and visits if no other tasks are feasible to be undertaken due to the lack of available range or payload. Each tour is defined as departing from the depot, undertaking at least one task, and returning to the depot. This MRTA problem is a class of combinatorial optimization problems, which are expressed generally in graph space. In this section, we express the MRTA problem as MDP over a graph. Then, MRTA is formulated as an optimization problem.

## 2.1 MDP over a Graph

The MRTA problem can be represented as a complete graph $\mathcal{G} = (V, E)$, which contains a set of nodes/vertices $(V)$ and a set of edges $(E)$ that connect the vertices to each other. Each node is a task, and each edge connects a pair of nodes. The weight of the edge $(\omega_{ij})$ represents the cost (e.g., distance) incurred by a robot to take task-$j$ after achieving task-$i$. For MRTA with $N$ tasks, the number of vertices and the number of edges are $N$ and $N(N-1)/2$, respectively. Node $i$ is assigned a 3-dimensional feature vector $d_i = [x_i, y_i, t_i]$.

In order to perform learning, the MRTA problem is defined as a Markov Decision Process (MDP) for each individual robot, which can be expressed as a tuple $< \mathcal{S}, \mathcal{A}, \mathcal{P}_a, \mathcal{R} >$. The components of the MDP are defined below. **State Space ($\mathcal{S}$):** A robot at its decision-making instance uses a state $s \in \mathcal{S}$, which contains the following information: 1) the current mission time, 2) its current location, 3) its remaining ferry-range (battery state), 4) its remaining payload, 5) the planned (allocated) tasks of its peers, 6) the remaining ferry-range of its peers, 7) the remaining payload of its peers, and 8) the states of tasks. The states of tasks contain the location, the time deadline, and the task status – active, completed, and missed (i.e., deadline is passed). **Action Space ($\mathcal{A}$):** The set of actions is represented as $\mathcal{A}$, where each action $a$ is defined as the index of selected task, $\{0, \ldots, N\}$ with the index of the depot as $0$. The task $0$ (the depot) can be selected by multiple robots, but the other tasks are allowed to be chosen once if they are active (not completed or missed tasks). $\mathcal{P}_a(s'|s, a)$**:** A robot by taking action $a$ at state $s$ reaches the next state $s'$ in a deterministic manner (i.e., deterministic transition model is defined). **Reward ($\mathcal{R}$):** The reward function is defined as $-f_{\text{cost}}$ and is calculated when there is no more active tasks (all tasks has been visited once irrespective of it being completed or missed). **Transition:** The transition is an event-based trigger. An event is defined as the condition that a robot reaches its selected task or visits the depot location.

## 2.2 MRTA as Optimization Problem

The exact solution of the MRTA problem can be obtained by formulating it as the following integer nonlinear programming problem:

$$\min f_{\text{cost}} = r - u(-r)e^{-d_r^2} \tag{1}$$

$$\texttt{subject to } \text{A set of integer linear constraints (Ghassemi et al. (2019))} \tag{2}$$

A detailed formulation of the exact ILP constraints that describe the MRTA problem with capacity and range restrictions, multi-tours per robot and tasks with deadlines, can be found in this recent work on decentralized MRTA (Ghassemi et al., 2019). Note that, in our paper we use a slightly different objective/cost function to better reflect the general problem setting presented here.

Here, we craft the objective function (Eq. (1)) such that it emphasises maximizing the completion rate (i.e., the number of completed tasks divided by the total number of tasks); and if perfect completion rate (100%) is feasible, then the traveled cost is also considered. In Eq. (1), $u(.)$ is the step function. The term of $1 - r$ is defined as task completion rate; i.e., the number of completed tasks ($N_{\text{success}}$) divided by the total tasks ($N$) or $r = \frac{N - N_{\text{success}}}{N}$. $d_r$ is a normalized value of the total distance traveled by all robots in the team. The term $d_r$ is the average travelled distance over all robots (i.e., $d_r = \frac{\sum_{i=1}^{N_r} d_i^{\text{total}}}{\sqrt{2} N}$). The terms $N_r$ and $d_i^{\text{total}}$ represent respectively the number of robots and the total traveled distance by robot-$i$ during the entire mission. The above objective function (Eq. (1)) gives positive value if the completion rate is lower than 100%, otherwise it gives negative value.

# 3 COVARIANT ATTENTION-BASED NEURAL ARCHITECTURE

Before describing the technical components of the proposed Covariant Attention Model, the so-called CAM neural architecture, we provide an illustration and summary description here of how this policy architecture is used by robots during an SR-ST operation. The CAM model for task allocation is called by/for each robot right when it reaches its current destination (task location or depot), in order to decide its next task or destination.

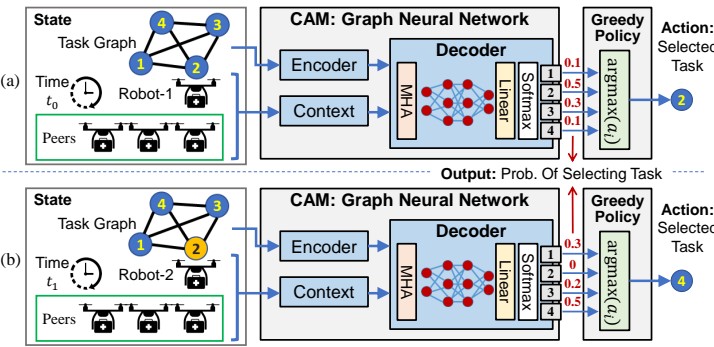

Figure 1: Deployment of the MRTA policy model based on the CAM architecture. a) Robot-1 at $t_0$. b) Robot-2 at $t_1$; here, the CAM output for previously selected task (e.g., task 2 in (b)) is set at 0.

Since full-observability is assumed across the multi-robot team and the policy-model execution-time is almost negligible, the current setup is agnostic to whether the online CAM model is executed centrally off-board or on-board each robot. As an example, Figure 1 illustrates how robot-1 and robot-2 uses the CAM policy model to choose a task at two different decision-making instances ($t = t_0$ and $t = t_1$, respectively). Here, the inputs to the CAM model includes **1)** the task graph information (i.e., the location of each task and its associated time deadline), **2)** the current mission time, **3)** the state of robot-$r$, and **4)** the states of robot-$r$'s peers. The CAM model then generates the probability of selecting each task as its output. A greedy policy to choose the task with the highest probability is used here, which thus provides the next destination to be visited by that robot. It should be noted that the probability values for completed tasks and missed tasks (i.e., missed deadline) are set at 0.

Figure 2 shows the detailed architecture of CAM. As shown in this figure, the CAM model consists of three key components, namely: Context, Encoder, and Decoder. The context includes the current mission time, the states of robot-$r$, and the states of robot-$r$'s peers. The encoder and decoder components are described below.

## 3.1 CCN-INSPIRED NODE ENCODER

The main function of the encoder is to represent the properties of each graph node and preserving its structural information into a continuous feature vector of dimension $d_{embed}$, which is digestible in the decoder. Each node $i$, has three properties which are the x-coordinate, y-coordinate, and the time deadline of the task. The encoding for each node should include its properties and the its positional association with its neighboring nodes. We implement a variation of CCN Hy et al. (2018). We determine the nearest $k$ (here we take $k = 5$) neighbors of a node ($Nb_i$) based on the positional coordinates ($x$ and $y$).

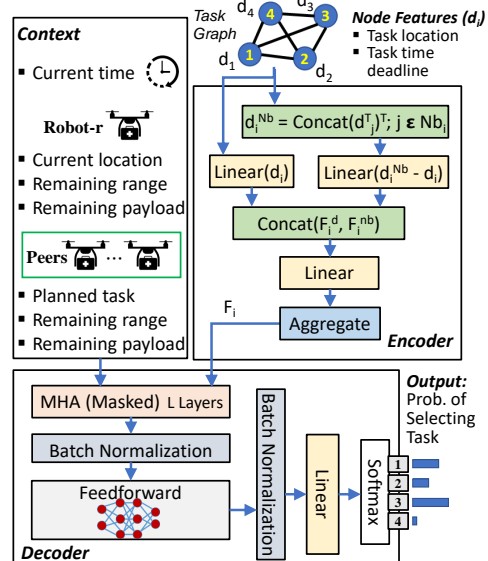

Figure 2: The architecture of the context, encoder and decoder for CAM.

The first step is to compute a feature vector by linear transformation for each node $i$.

To encode the node properties, we do a linear transformation of $d_i$ to get a feature vector $F_i^d$ for all $i \in [1, N]$, as shown in Eq. (3).

$$F_i^d = W^d d_i^T + b_d \tag{3}$$

where $W^d \varepsilon \mathbb{R}^{d_{embed} \times 3}$, $b_d \varepsilon \mathbb{R}^{d_{embed} \times 1}$. For effective decision making, we also need to preserve the structural information. Therefore we define a matrix $d_i^{Nb}$ as defined in Eq. (4).

$$d_i^{Nb} = \text{Concat}(d_j^T), \ j \in Nb_i \tag{4}$$

We do a linear transformation of $d_i^{Nb}$ to get $F_i^{Nb}$ (as shown in Eq. (5)), which we believe captures the closeness of a node with its neighbours in terms of the node properties.

$$F_i^{Nb} = W^{Nb}(d_i^{Nb} - d_i^T) + b^{Nb} \tag{5}$$

where $W^{Nb} \; \varepsilon \; \mathbb{R}^{d_{\text{embed}} \times 3}$, $b^{Nb} \; \varepsilon \; \mathbb{R}^{d_{\text{embed}} \times 1}$. $F_i^{Nb}$ captures the information about how close the node properties of neighbor nodes of node $i$ is to itself, which shows a representation of how important node $i$ is to its neighbors. We compute the final embedding for each node using Eq. (6).

$$F_i = \text{Aggregate}(W_f(\text{Concat}(F_i^d, F_i^{Nb})) + b_f) \tag{6}$$

Here, $W_f \; \varepsilon \; \mathbb{R}^{d_{\text{embed}} \times d_{\text{embed}}}$, $b_f \; \varepsilon \; \mathbb{R}^{d_{embed} \times 1}$. Thus finally we get an embedding $F_i$ for each node, where $F_i \; \varepsilon \; \mathbb{R}^{d_{\text{embed}} \times 1}$. $W^d$, $b_d$, $W^{Nb}$, $b^{Nb}$, $W_f$, and $b_f$ are learnable weights and biases. The Aggregate function here is the summation across all the columns of a matrix. This summation along with the relative difference in node properties, as in Eq. (5), preserves permutation-invariance and the structural properties (cognizance of inter-node distances for example) of the graph. Note that, these operations make the encoded state (w.r.t. a given node) insensitive to the order of the neighboring nodes, and thus the overall state space becomes independent of the indexing of tasks or to rotations of the graph. These permutation-invariance characteristics are important for the consistency of the state space encoding, and further helps in the generalizability of the learnt policies (which should be cognizant of the structure of the task-space graph) over unseen scenarios.

## 3.2 ATTENTION-BASED DECODER

Kool et al. (2019) presented an attention mechanism-based model for solving combinatorial optimization problem, where an attention-based encoder and decoder are implemented. Attention mechanism became popular because of it successful implementation in natural language processing. The intuition behind attention mechanism, is to focus on the most relevant part (in terms of language translation). The same concept can be applied for a combinatorial optimization problem with sequential decision making.

The main objective of the decoder is to use the information from the encoder, and the current state as context, choose the best task by calculating a probability value for each node. The first step is feeding the output from the encoder (as key-values) and information from the current state (as context) to a multi-head attention (MHA) layer. The context for the MHA layer in this experiment consist of the following seven features: 1) Current time; 2) Available range of the robot taking decision; 3) Current location of robot taking decision; 4) Current payload delivery capacity; 5) Current destination of other robots; 6) Available range for other robots; and 7) Current payload delivery capacity for other robots. Figure 2 illustrates the structure of the decoder. Attention mechanism can be described as mapping a query $(Q)$ to a set of key-value $(K, V)$ pairs. The inputs, which are the query $(Q)$, key $(K)$, and values $(V)$, are all vectors. The output is a weighted sum of the values $V$, weight vector is calculated by the compatibility function:

$$\text{Attention}(Q, K, V) = \text{softmax}(QK^T/\sqrt{d^k})V \tag{7}$$

where $d^k$ is the dimension of $K$ or $V$. In this work we implement a multi-head attention (MHA) layer in order to determine the compatibility of $Q$ with $K$ and $V$. The MHA implemented in this work is the same as the one implemented in (Kool et al., 2019) and (Vaswani et al., 2017). As shown in (Vaswani et al., 2017) the MHA layer can be defined as:

$$\text{Multihead}(Q, K, V) = \text{Linear}(\text{Concat}(\text{head}_1 \dots \text{head}_h)) \tag{8}$$

where $\text{head}_i = \text{Attention}(Q, K, V)$ and $h$ is the number of heads. The feed-forward layer is to convert the output from the MHA layer to a dimension that can be taken in for the next process. The final softmax layer outputs the probability values for all the nodes. The next task to be done will be chosen based on a greedy approach, which means the node which has the highest probability will be chosen. The nodes which are already visited will be masked such that it will not be chosen in the future time steps of the simulation.

## 4 LEARNING FRAMEWORK

Both the CCN-inspired encoder and the attention-based decoder consist of learnable weight matrices as explained in Sections 3.1 and 3.2. In order to learn these weight matrices, both supervised and unsupervised learning methods can be used. However, supervised learning methods are not tractable

since the computational complexity of the exact I(N)LP solution process required to generate labels. The complexity of the ILP formulation of the MRTA problem scales with $O(n^3 m^2 h^2)$, where $n$, $m$, and $h$ represent the number of tasks, the number of robots, and the maximum number of tours per robot, respectively Therefore, we use a reinforcement learning algorithm to conduct the learning.

**Learning Method:** In this work we implement a simple policy gradient method (REINFORCE) as the learning algorithm with greedy rollout baseline, which also enables us to compare the effectiveness of our method with that of (Kool et al., 2019). For each epoch, two sets of data are used which are the training set and the validation set. The training data set is used to train the training model ($\theta_{\text{CAM}}$) while the validation data set is used to update the baseline model ($\theta_{\text{CAM}}^{BL}$). The size of the training data and the validation data used for this paper is mentioned in Section 5.1. Each sample data from the training and validation data-set consist of a graph as defined in Section 2.1. The pseudo code of the training algorithm for our architecture is shown in Alg. 1. It should be noted that the policy gradient method requires the evaluation of a cost function, which is defined to be same as in Eq. (1). **Policy:** We define the policy such that if the robot $r$ does not satisfy the constraints in Eqs. (2), it returns to depot (i.e., $a = 0$). Otherwise the robot $r$ runs the learnt CAM network and chooses the output (task) based on a greedy approach (selects a task with the highest probability value), as shown in Fig. 1.

---

**Algorithm 1** CAM - Training Algorithm

**Inputs:** $N_{\text{epoch}}$: # of epochs, $B$: Batch size, $N_{\text{tr}}$: Training data size, $N_{\text{vl}}$: Validation data size; $\theta_{\text{CAM}}$: CAM model; $\theta_{\text{CAM}}^{BL}$: Baseline CAM model.

1: **for** epoch = 1..$N_{\text{epoch}}$ **do**
2:     $\mathcal{D}_{\text{tr}}, \mathcal{D}_{\text{vl}} \leftarrow \text{GenerateScenarios}(N_{\text{tr}}, N_{\text{vl}})$
3:     **for** step = 1..$\lfloor N_{\text{tr}}/B \rfloor$ **do**
4:         $\mathcal{D}_{\text{tr,b}} \leftarrow \text{RandSample}(\mathcal{D}_{\text{tr}}, M)$
5:         $\mathbf{a}^{\text{BL}}, f_{\text{cost}}^{\text{BL}} \leftarrow \text{CalCost}(\theta_{\text{CAM}}^{BL}, \mathcal{D}_{\text{tr,b}})$
6:         $\mathbf{a}, f_{\text{cost}} \leftarrow \text{CalCost}(\theta_{\text{CAM}}, \mathcal{D}_{\text{tr,b}})$
7:         $\nabla\mathcal{L} \leftarrow \frac{1}{B}\sum_{i=1}^{B}(f_{\text{cost,i}} - f_{\text{cost,i}}^{\text{BL}})\log\text{softmax}(\mathbf{a}_i)$
8:         $\theta_{\text{CAM}} \leftarrow \text{ADAM}(\nabla\mathcal{L}, \theta_{\text{CAM}})$
9:     $\mathbf{a}_{\text{vl}}^{\text{BL}}, f_{\text{cost,vl}}^{\text{BL}} \leftarrow \text{CalCost}(\theta_{\text{CAM}}^{BL}, \mathcal{D}_{\text{vl}})$
10:     $\mathbf{a}, f_{\text{cost}} \leftarrow \text{CalCost}(\theta_{\text{CAM}}, \mathcal{D}_{\text{vl}})$
11:     **if** $(\sum_{i=1}^{N_{\text{vl}}} f_{\text{cost,i}}^{\text{BL}} > \sum_{i=1}^{N_{\text{vl}}} f_{\text{cost,i}}) \wedge (\text{T-Test}(\mathbf{a}_{\text{vl}}, \mathbf{a}_{\text{vl}}^{\text{BL}}) > \epsilon)$ **then**
12:         $\theta_{\text{CAM}}^{BL} \leftarrow \theta_{\text{CAM}}$
13: **procedure** CALCOST($\theta, \mathcal{D}$)
14:     $\mathbf{a}, f_{\text{cost}} \leftarrow [\,], [\,]$
15:     **for** $i = 1..|\mathcal{D}|$ **do**
16:         $\mathbf{a}_i, f_{\text{cost,i}} \leftarrow \text{Simulation}(\theta, \mathcal{D}_i)$
17:         $\mathbf{a} \leftarrow \mathbf{a} \cup \mathbf{a}_i$
18:         $f_{\text{cost}} \leftarrow f_{\text{cost}} \cup f_{\text{cost,i}}$
19:     **return** $\mathbf{a}, f_{\text{cost}}$

---

## 5 CASE STUDIES

In this paper, we consider a flood disaster response application, where a team of UAVs deliver survival kits to flood victims. Here we assume that the information on the locations of victims (tasks) is available a priori. We design and execute a set of numerical experiments to investigate the performance of our proposed learning-based algorithm over graph space (*CAM*) and compare it with an extended version of a state-of-the-art graph learning-based algorithm proposed by Kool et al. (2019), so called *attention-based* (*AM*) approach. In order to test and validate the approaches, a set of numerical experiments is designed to mimic flood scenarios. The results are compared in terms of the cost function (Eq. (1)), and completion rate.

### 5.1 DESIGN OF EXPERIMENTS & LEARNING PROCEDURES

To evaluate the proposed CAM method, we define an MRTA case study with 20 UAVs and 200 task (flood victims) locations. A 2D environment with 1 sq. km area is used for this purpose, with the time deadline of tasks varied from 0.1 to 1 hour. A randomized sample case scenario is shown in Fig. 3, which depicts the task locations, depot location and task time deadlines. The UAVs are assumed to have a range of 4 km, a maximum capacity of 10 payloads and a nominal speed of 10 km/h. We assume instantaneous battery swap and payload replenishment are provided at depot location, which is used when UAVs return to depot since they have either exhausted their (survival kit) payloads or were running low on battery. It is important to note that the flood victim application is used here merely for motivation, and the CAM architecture is in no way restricted to this application, but can rather solve problems in the broad (important) class of capacity/range-constrained and timed task-

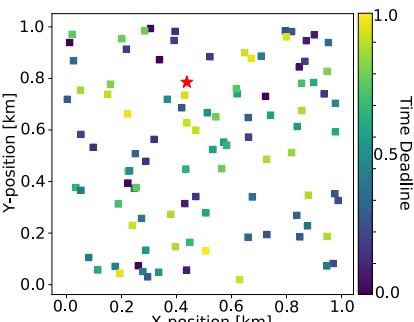

Figure 3: Sample scenario showing depot location (red star) and task locations (squares) colored by their time deadlines.

constrained SR-ST problems. Moreover, even the policies learnt here for CAM demonstration on the described case settings can generalize to related SR-ST problems with up to 200 tasks, which represents a fairly large MRTA problem in reference to the existing literature in the MR domain.

To perform learning and testing of the learned model, we proceed as follows: *Learning Phase:* We use a policy gradient reinforcement learning algorithm (REINFORCE with rollout baselines in this case) for learning the optimal policy. The learnable parameters in this architecture includes all the weights in the encoder and the decoder. The training is carried out for a total of 200 epochs. Each epoch consists of 10,000 random training samples, which are evaluated and trained in batches of 100 samples. *Testing Phase:* In order to provide a statistically insightful evaluation and comparison, 100 random test scenarios are generated for the 20-uav-200-task case study, such that the number of flood victims are constant (200) but their locations and time deadlines are different.

### 5.2 MODIFICATIONS TO AM

The attention-based mechanism (AM) reported by Kool et al. (2019) has been shown to solve a few different classes of single-agent/robot combinatorial optimization problems. To be able to implement the AM method for our problem (for comparison with our CAM method), the AM method is adapted to a multi-robot setting. For this purpose, we make the following three changes to the AM method: (i) The node properties that are defined in Section 2.1 are used in AM; (ii) The context for the attention mechanism is modified to be the same as that used for CAM; and (iii) The cost function used for training is changed to that in Eq. (1).

### 5.3 SIMULATION AND FRAMEWORK SETTINGS

The *"Python"* 3.7 and the 64-bit distribution of *"Anaconda 2020.02"* are used to implement the MRTA approaches. The environment, CAM architecture, training algorithm, and the evaluation of the trained model, are all implemented in *Pytorch-1.5*. The attention-based architecture was also implemented in Pytorch. With the help of Pytorch, the training was deployed on two GPUs (NVIDIA tesla v100-pcie-16gb) with 16 GB RAM.

## 6 RESULTS AND DISCUSSION

**Learning Curve:** In order to compare the sample efficiency of the proposed CAM method with that of the AM approach, we run both methods with similar settings and plot their learning curve (convergence history), as shown in Fig. 4. As seen from this figure, our CAM method converges faster and to a significantly better (by 69.4%) optimal cost function value ($f^*_{\text{cost,CAM}} = 0.029$) compared to AM ($f^*_{\text{cost,AM}} = 0.093$). In contrast to the 73 epochs

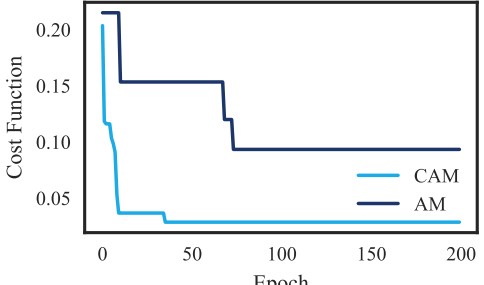

Figure 4: Convergence plot comparing CAM and AM for 200 epochs

taken by AM to converge, CAM reached the best cost value of AM only after 7 epochs and reached its final converged value in 35 epochs. These results clearly demonstrate the better sample efficiency of our proposed method, CAM, in comparison to AM. As described later the computational cost of each epoch for our CAM method is also ∼2 minutes smaller than that of the AM method. Overall, our proposed approach reaches better cost function values significantly faster in terms of number of training epochs, with the benefits being more pronounced in terms of wall time. The higher cost of AM could be in part attributed to directly implementing the encoding architecture of a transformer network (Vaswani et al. (2017)); this transformer mechanism was designed for machine translation and thus consists of multiple layers of Multi-head attention computations with a feed-forward structure. In contrast, our CAM model uses simple linear transformations of the node properties and its relative differences in local neighborhoods to capture structural information. Thus the total number of learnable parameters in AM (almost 393600 weights) is significantly larger than that in our CAM architecture (almost 17400 weights), leading to slower convergence of the former.

**Real-time Performance:** Based on the epoch information in Section 5.1, the average time to complete a training epoch was found to be 12.48 min for AM and 10.17 min for CAM. The average decision making time of both methods are also comparable at ∼2.3-2.4 ms.

**Generalizability Analysis:** Figure 5 shows the performance of the CAM and AM approaches for 100 unseen test scenarios in terms of cost function (the lower the better) and completion rate (the

higher the better). It can be seen from Fig. 5 that our proposed CAM method outperforms the AM approach by achieving 84% and 5% better median values of the cost function and the completion rate, respectively. We posit that the generalizability benefits of the CAM architecture can be attributed to its covariant compositional encoding, which is able to aggregate local node neighborhoods while remaining agnostic to node ordering. The local structure of the graph is not only important to effective decision-making, but also expected to be shared across various problems settings drawn from the same distribution, thereby promoting generalizability of policies when adequately captured.

**Scalability Analysis:** In order to study how the performance of the learnt model is generalizable to new unseen scenarios with different different task-space size (50 to 200 tasks), a set of tests are conducted with the 20-robot swarm. For each task size, 100 random scenarios are generated and tested. The results are shown in Fig. 6. In this figure, the dashed horizontal line shows the threshold value of 0, where any cost below that line depicts a perfect completion rate (100%); when imperfect, the net traveled distance is taken into consideration. The performance of both CAM and AM approaches improved by decreasing the number of total tasks. It can be seen from Fig. 6 that

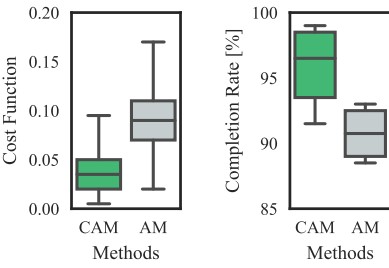

(a) Cost function     (b) Completion rate

Figure 5: Generalizability analysis: The performance of the algorithms over 100 unseen test case scenarios. Each scenario includes 200 tasks and the swarm size is set at 20 robots.

our primary method, CAM, outperforms the AM approach in all case scenarios. The proposed CAM method provided a 100% completion rate for 50-, 100-, and 150-task cases, while the AM method provides a 100% completion rate for 50- and 100-task cases. In terms of the cost function, the CAM method achieves up to 435% better values compared to the AM method.

# 7 CONCLUSION

In this paper, we proposed a new graph neural network architecture, called CAM, for a multi-robot task allocation problem with a set of complexities, including tasks with time deadline and robots with ferry range and payload capacity constraints. This new architecture incorporates an encoder based on covariant node-based embedding and a decoder based on attention mechanism. To learn the features of the encoder and decoder, the problem has been imposed as a reinforcement learning problem and it has been solved using a policy simple gradient algorithm, REINFORCE. In addition, to compare the performance of the proposed CAM method, an attention-based approach (aka AM) has been extended to be able to handle a multi-agent combinatorial optimization problem (i.e., a multi-robot task allocation problem).

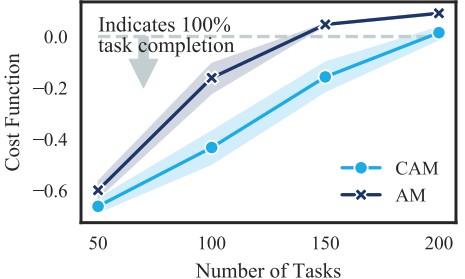

Figure 6: Varying task size for a 20-robot swarm: The performance of the CAM, and AM algorithms over varying task size for 100 unseen test case scenarios per task size. The solid line shows the mean value and the shaded area shows the 95% confidence interval. The dashed gray line shows the threshold value of 0, where any cost below of this line means a 100% completion rate.

To evaluate the performance of the proposed architecture, and the extended version of AM are trained for 200 epochs and tested on 100 unseen case studies. Performance was analyzed in terms of cost function and completion rate. The new proposed architecture showed a better sample efficiency than AM by reaching a better cost value only after 7 epochs versus 73 epochs of AM. Our primary method, CAM, outperformed the AM approach by achieving (up to) 84% better cost function value (in term of the mean value). The computational cost analysis showed that the proposed CAM model takes a few milliseconds to take a decision; hence, it is an excellent choice for online decision making (here, task allocation). A further study on the performance of both AM and CAM approaches on case studies with different number of tasks demonstrated that our proposed CAM method is superior to AM in all case studies. While the current method is operational over varying task size (the upper bound is the number of tasks that model has been trained on), it is not invariant to the size of swarm. One immediate future work is to expend the architecture to have a swarm size invariant model.

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
