# OpenReview forum: "Learning to Solve Multi-Robot Task Allocation with a Covariant-Attention based Neural Architecture"
_ICLR.cc/2021/Conference — Reject_

### Official Review · AnonReviewer3 · 2020-10-15
**Interesting idea, but more clarity and more thorough experimentation are needed**

**Rating:** 4
**Confidence:** 4

**Review:**

This paper presents an interesting idea of using neural-network-based RL to solve a type of vehicle routing problems, where the vehicles are tasked with visiting spatial locations to deliver items, and are subject to load capacity and delivery time constraints. In order to solve this problem, the authors propose an encoder-decoder architecture to decide for each robot, where to move next. The encoder is inspired from the Covariance Compositional Networks and the decoder utilizes an attention module, and the network is trained via REINFORCE. The results show that the proposed method outperforms the baseline in unseen test cases, in terms of task completion rate and the specified cost function.


Overall, the idea of the paper is interesting. It extends the work by Kool et al. 2019 to incorporate constraints in the optimization problem, such as payload capacity and visit deadlines. However, the paper could be improved in terms of clarity and mathematical rigor. Moreover, the reviewer expects more extensive experiments to demonstrate the efficacy of the proposed method, as well as discussion on what enables the proposed method/network to work better). As such, the reviewer believes that this paper is not suitable for publication in its current form. However, the idea of extending the existing method to take into account various constraints is novel and useful, and the reviewer believes that an improved version of this paper would make a good contribution in the area of vehicle routing optimization.


Here are the detailed comments.
1. Contributions

$\quad$ It seems that in the paper, the authors are targeting a specific problem scenario: A team of robots are tasked with delivering items to some spatial locations, with load capacity constraints for the robots and delivery deadlines for the locations. This is an interesting scenario, but seems like a specific application, which makes the contribution of this paper a bit narrow. The reviewer believes that the proposed learning-based approach could actually be applied to a wider range of "standard" vehicle routing problems with temporal constraints, such as the orienteering problem with time window/deadlines, the TSP with time window/deadline, the prize-collecting TSP with temporal constraints, some of which are discussed in Nunes et al. 2017, in addition to the problem studied in the paper. The reviewer encourages the authors to consider these possibilities and also compare with the state-of-the-art optimization-based and learning-based methods for some of these standard problems.


2. Clarity

There are many things that could have been explained more clearly. Here are a few examples.

$\quad$ a. There should be a formal definition of the optimization problem, which shows the objective function, the constraints, and the variables to be optimized. The authors should also explain what the solution looks like, e.g., it's a sequence of locations to be visited for each robot. Although such information might be pieced together from different places in the paper, it'd be of a great value to present a formal optimization formulation.

$\quad$ b. There are no clear descriptions of what the obtained decision rule is like. While it could be understood from the network architecture and descriptions here and there that each robot runs the network to obtain the next location (unless it runs out of battery and/or items), a clear algorithmic description of the decision rule should be provided.

$\quad$ c. In Sec. 2.3, for the context part, how are the current destinations of the other robots determined? It seems that each robot will need to wait for the other robots to compute their next destinations, before it can compute its own. But then how does one resolve the cyclic dependencies here?

$\quad$ d. There should be a discussion on what enables the proposed method to outperform the baseline, in terms of training speed and test performance.

$\quad$ e. The authors could provide an intuitive explanation about the permutation invariance in Sec. 2.3, in the context of this routing problem.

$\quad$ f. In the results section, it is not clear how the approach of Kool et al. 2019 is modified for the problem in this paper. What are the changes, and is it trained using the cost function of Eq. 9 or the original cost function in Kool et al. 2019?

$\quad$ g.  The introduction of the problem type at the beginning of page 2 is confusing. It should actually be multi-robot task and single-task robots (based on Nunes et al. 2017).

$\quad$ h. In the sentence above Sec. 1.1, what does multi-scale features mean here? What are the scales in the context of the routing problem in this paper?


3. Mathematical rigor

$\quad$ a. The matrix equation in Eq. 3 is problematic. The dimensions of $W$ and $d$ don't match. The authors probably want to left multiply $d$ with $W$.

$\quad$ b. In Eq. 5, the dimensions of $d_i^{Nb}$ and $d_i$ don't match and the subtraction cannot be performed. Although it is understood what operation the authors want to do here, this should be written properly.

4. Other comments

$\quad$ a. Typos. Here are just a few examples. Please check through the paper.

 $\quad$ $\quad$  page 1, salesmn -> salesman

  $\quad$ $\quad$ page 2, learn solve -> learn to solve

 $\quad$ $\quad$  page 3, extra hyphen between in and order

  $\quad$ $\quad$ page 4, therefor -> therefore

$\quad$ b. Not properly-written sentences

  $\quad$ $\quad$ page 3, paragraph above Sec. 1.3, "this can be used to..." -> "which can be used to..."

  $\quad$ $\quad$ page 4, "Therefor we define a matrix $d_i^{Nb}$ as defined in Eq. 4."

$\quad$ c. In page 4, the notation for matrix/vector dimension is a bit strange, or at least the authors should define it. The authors could have used the standard way of "$W \in \mathbb{R}^{m\times n}$".

$\quad$  d. Fig. 2 and 3 are not properly placed and overlap with the text.

---

> ### Author Response · Authors · 2020-11-24
> **2a. Formal optimization objective and MDP formulation added and reference for the formal definition of the exact optimization constraints provided.**
>
> As described in our other responses, we have now included a new section, Section 2, which clearly explains the Optimization and MDP formulations of the class of MRTA problems being tackled here. Note that, with regards to the "optimization formulation", we mainly provide detailed formulation/description of the objective function (Section 2.2), and refer readers to another paper (Ghassemi et al., 2019, https://arxiv.org/pdf/1907.04394.pdf) that presents the exact formulation of the many constraints involved in this problem. To save space, the long list of these constraints is not re-stated in our paper.
>
> In addition, the new diagram (Fig. 1) added in the revised manuscript, along with the added formal definition of the MDP problem (Section 2.1) now provides a clearer visualization/description of what the task-allocation solution (or the actions of the policy model) looks like for each robot in the multi-robot team. Note that, in its current form, every time the CAM model is triggered, it yields the next location to be visited by a given robot (i.e., next task selected by it), thus demonstrating a myopic implementation in its current form (now explained in Section 2, first paragraph). The outcome of instantiating the CAM model after every task leads to the sequence of tasks performed by the robot during the overall operation.

---

> ### Author Response · Authors · 2020-11-24
> **2b. Execution of the learned policy model explained, augmented with a new diagram in the revision.**
>
> We agree that the original manuscript did not explicitly demonstrate how the decision-making works. As explained in more detail in our responses to Reviewer 2 comments, we have now added a new section (Section 2) that clearly describes the MRTA problem as an optimization problem and a Markov Decision Process (MDP). Moreover, at the start of Section 3, we have added a new paragraph and a diagram (Fig. 1) to clearly describe how the learnt CAM model works. As an example in the revised manuscript, Fig.~1 illustrates how robot-1 and robot-2 use the CAM policy model to choose a task at two different decision-making instances ($t=t_0$ and $t=t_1$, respectively). The decision rules guiding task allocation are now readily evident from these additions made to the manuscript.

---

> ### Author Response · Authors · 2020-11-24
> **2c. Conflict resolution and mitigation of cyclic dependencies clarified.**
>
> In this paper, we assumed full observability (i.e., perfect communication). To paraphrase, the locations and time deadline of all tasks are assumed to be known by all robots. Additionally, at each decision-making instance, robot-$r$ uses its most recent information that has been received from the other robots to determine the state space and context. When a robot takes a decision, it immediately broadcasts its decision to all other robots. This decision is depicted as the index of the selected task.
>
> Theoretically speaking, the dependency on other robots' decisions can cause a cyclic dependency in the case of conflicts. However considering that decision times (with a learnt model) and communication latency (given the minuscule data being shared) are orders of magnitude smaller than travel times of robots, the likelihood of multiple robots taking decisions simultaneously was found to be quite low (or negligible) in our simulations. In addition, as a practical fail-safe, we consider a simple priority-based rule, where a robot with a lower id has a higher priority of selection, in the rare cases of decision conflicts. We agree that this issue might become prominent in very large swarm robotic operations, which remain an open area of work in both RL and non-learning based MRTA domains.

---

> ### Author Response · Authors · 2020-11-24
> **2d. Insights provided on how CAM results in better performance over the baseline attention mechanism (AM).**
>
> Short discussions on what enables the CAM method to outperform the AM method are now provided in the various results analysis portions of Section 6 in the revised manuscript, and highlighted in blue font.

---

> ### Author Response · Authors · 2020-11-24
> **2e. Insights provided on how the permutation invariance helps in generalizing the MRTA decision-making.**
>
> A brief explanation has been added at the end of Sec. 3.1 w.r.t. how permutation invariance helps in the MRTA problem. In Eq.(6), note that the aggregate function is basically a summation across all the columns of a matrix, which means the relative importance of a node with respect to its neighbors is aggregated into one vector irrespective of the order of the neighbors. In addition, Eq. 5 allows accounting for the distance between nodes, thus helping preserve the physical structure of the graph. Moreover, any change in the indexing of tasks, and/or rotation of the graph, will cause no change in the CCN embedded graph information, thus allowing the policy network to encode the state space consistently and better generalize (the routing decisions) across different scenarios.

---

> ### Author Response · Authors · 2020-11-24
> **2f. Adaptation of the AM method (to MRTA) for comparison now clearly described in revision.**
>
> As stated in one of our earlier responses to Reviewer 2, we have now added a small section (Section 5.3 in the revised manuscript), to clearly describe how we adapt the baseline AM method (by Kool et al. 2019) to the specific class of MRTA problems studied here, for comparison with our CAM method. Note that both the AM and our CAM methods are trained on the same cost function, now given in Eq. 1 of the revised manuscript.

---

> ### Author Response · Authors · 2020-11-24
> **2g. Exact class of MRTA problems studied here (which is SR-ST) now clarified.**
>
> We thank the reviewer to help us clearly point out the class of MRTA problems being studied here. In this paper, we assume each task only requires one robot to be accomplished (i.e., Single-robot Task) and each robot can only perform one task at a given time (i.e., Single-task Robot). Therefore, the class of our problem is SR-ST. We have now clearly stated this at the start of the paper, in Section 1 (at the beginning of page 2).

---

> ### Author Response · Authors · 2020-11-24
> **2h. Loose usage of the "multi-scale" term now removed.**
>
> The term "multi-scale" was somewhat loosely used in the original manuscript. What we instead intended to say is that the present architecture can capture and preserve local structural information of the graph over which learning occurs, and this ability along with implicit tensor representations could be helpful in the case of multi-level graphs that includes multiple node properties. The term "multi-scale" has now been removed from the introduction section.

---

> ### Author Response · Authors · 2020-11-24
> **3a, 3b, 4b, 4c. Concerned equations addressed.**
>
> The concerned equations have now been updated in the revised manuscript to address the above-stated issues.

---

> ### Author Response · Authors · 2020-11-24
> **4a. Typos fixed.**
>
> We are grateful to the reviewer for pointing out these typos and issues. Suggested corrections have now been carefully implemented throughout the revised manuscript.

---

> ### Author Response · Authors · 2020-11-24
> **Broader technical contribution clarified. Explained how the presented method CAM can readily solve various other mTSP and VRP problems with temporal constraints..**
>
> Based on the MDP formulation of our problem (now added as Section 2.1 in our revised manuscript), we can safely say that CAM can readily translate to solving most of the other TSP and VRP problems in Nunes et al. 2017; and thus our contributions are not limited to the "seemingly" narrow application scope of the multi-UAV flood response case study used (we used this case study with the goal to provide practical context to the stated complexities that are important to such MRTA/SR-ST problems in general). Moreover, given that the methodological advantage of CAM is likely attributed to its encoder (with training efficiency partly attributed to our adaptations of the attention mechanism), where this encoder is able to capture local structure of the graph and remain invariant to node ordering, we expect the advantage to translate to other related classes of problems where these characteristics are important. The referenced mTSP and VRP problems fall into such classes (often involving simpler state/action spaces comared to our MRTA problem), and hence are expected to benefit from the CAM architecture. While generating and adding the results of CAM on these additional benchmark problems was practically challenging (purely from a time perspective) within this short 1.5 week revision period, we are fairly confident of our above-stated claims, and can readily add these results in the final manuscript, if our paper is accepted.
>
> For example, the Solomon's benchmark problem, VRP with Time Window (VRP-TW) in Nunes et al. 2017 is similar to our studied MRTA problem except that no payload/range constraints are present and the task properties also include a minimum start time. To this problem, one can readily apply our CAM method by using the cost function as stated in that benchmark, and making the following minor changes to our state space:
> State space: REMOVE 1) remaining payload of robot-r, 2) remaining range of robot-r, 3) remaining payload of robot-r's peers, and 4) remaining range of robot-r's peers. These would reduce the inputs to the context portion of the CAM architecture. ADD 1) minimum starting time for each task. This would become a fourth property in our node feature vector (d) corresponding to each task, which is fed as an input to the encoder portion of CAM. The action space will remain the same, except that tasks whose "minimum-starting-time > current-time + time-of-travel-to-task" wont be available to be selected; in our architecture the probabilities of these tasks can be set as zero by masking the corresponding outputs, similar to how we currently deal with missed tasks.
>
> To reiterate, this paper's objective is not focussed on just solving the multi-UAV flood response application. The objective of this paper is to develop and demonstrate the new neural architecture CAM (involving the novel encoder and the modified decoder and context) for solving MRTA or mTSP/VRP problems with agent capacity and task-temporal constraints, though an RL approach.

---

### Official Review · AnonReviewer4 · 2020-10-22
**An adaptation of an existing method to a particular domain, lacks significance**

**Rating:** 5
**Confidence:** 4

**Review:**

Summary
-------
The authors adapt an existing RL approach to combinatorial optimization to be used for their particular application of optimizing a fleet of UAVs (simulation) to deliver supplies. For this the problem is presented as a graph, a cost function is defined and an optimization is applied.


Critique, Questions, and Discussion
-----------------------

1) In order to follow the explanation better I would recommend describing the task in more detail in 2.1: What are the actions? What is the exact optimization problem that is being solved? Figure 1 shows that the outputs of the architecture are the action probabilities, but the reader still does not know what the actions are and has no clear picture of the problem that is being solved.

2) Turns out the paper never mentions what the actions are and does not specify the MDP to which the reinforcement learning approach is being applied.

3) It is not explained how the competitor methods (AM) is adapted to the problem at hand.

4) I would recommend to clearly state somewhere in the beginning of the paper what are the specific contributions of this work from methodological standpoint.

5) It is not clear from the description of the task why formulating the problem as a graph is beneficial in this scenario. How does the distance information, for example, enter the decision-making of the agents? Maybe a clearer explanation of the task and the exact optimization problem would help the reader see it.

6) How efficient would be a classical (non-RL) solution on this particular problem? It would be interesting to see some numerical estimation of this that would explain the necessity for heuristics in this problem.


Recommendation and justification
--------------------------------
In my opinion the potential impact of this paper is not sufficient (even if perfectly executed) to be presented at ICLR. This work does not offer methodological novelties, and the particular application is of limited significance as the experiments are conducted on a toy problem, not on the real task that is given as motivation. To the best of my understanding this work is interesting from engineering standpoint as application of RL-based combinatorial optimization to a particular problem, but does not constitute scientific contribution.


Additional remarks
------------------
Typos
pg 3: "consist of multiple tasks located in different locationS"
pg 3: "Therefore its possible" -> it's
pg 4: "ThereforE we define a matrix"


--------------------- UPDATE Nov 30 ---------------------
-----------------------------------------------------------------

I find the updated manuscript to be a significant improvement over the initial version. The class of problems is now clearly described and the problem formulation explains the challenge and the need for an RL-based solution approximation.

Seeing now that this work is not about a particular deployment scenario, but rather aims to present a method that can be used to solve a general class of constrained routing problems, another set of question arise.

Probably the main limitation that precludes from evaluating the significance of contribution and seeing this work as general contribution to the list of solution for routing problem is the fact that is was evaluation on only single instance of such problem, with only single set of experiments parameters (20 robots / 200 tasks). Since the paper aims to propose a general method that is widely applicable is it only reasonable to expect to see experimental evidence that when applied to a diverse set of problems of this class it comes out on top. How does it fare against other similar routing optimization problems? How does if fare against other methods or AM is the only method that is worth comparing to?

With this in mind I am raising my score for this paper from 3 to 5 "Marginally below acceptance threshold". The reason for not going higher is the lack of demonstration of applicability of this methods to a wider range of problems.

---

> ### Author Response · Authors · 2020-11-24
> **Clarifying methodological novelty, and addressing the characterization of the paper as "an adaptation of existing methods to new domain"**
>
> The statement that the presented method is "An adaptation of an existing method to a particular domain" is not an adequate characterization of the work; we take the responsibility for the lack of clarity in the original manuscript that might have led to this characterization. To paraphrase, we agree that the original manuscript did not do a satisfactory job of clearly pointing out the contributions, and of explaining the general class of multi-robot task allocation (MRTA) problems being formulated as an MDP over graph, and solved here. These issues have now been addressed in the revised manuscript, as further described in the other responses here. To summarily clarify -- in this paper we present an overall new neural architecture, called covariant attention model or CAM, for learning policies to solve an important class of MRTA problems, namely Single-task Robots and Single-robot Tasks (SR-ST) problems with range/capacity constraints on robots and tasks with deadlines, as opposed to merely addressing a particular application. The multi-UAV flood-response application setup is simply used to evaluate the work, and demonstrate potential real-world applicability. Firstly, we now present the formal Markov Decision Process formulation of the concerned class of MRTA problems. More importantly, akin to many other work in this domain of learning over graphs, we not only modify existing architectures (in our case, Attention Mechanism or AM) to construct the decoder and context layers, but also formulate and integrate a new type of encoder based on covariant compositional layers. More specifically, the encoder is our contribution in the context of solving MDP over graphs, and the results indicate that the main performance gains provided by our CAM method (over the baseline attention mechanism architecture) is mostly attributed to this new encoder.

---

> ### Author Response · Authors · 2020-11-24
> **1. Optimization and MDP formulations of the studied class of MRTA problems now clearly stated in revised manuscript.**
>
> We agree that the original manuscript did not provide: i) sufficient mathematical description of the general task allocation (SR-ST) problem being addressed here, and ii) clear description of how the CAM model works. These issues have now been addressed, as summarized below.
>
> 1. Problem Definition: A whole new section (as Section 2) has been added in the revised manuscript to clearly define the problem both in terms of an optimization problem (essentially an integer non-linear programming or INLP problem) and as a Markov Decision Process (MDP) over graphs. Note that, with regards to the optimization problem definition, we mainly state the mathematical objective function here, and refer readers to another paper (Eqs. 3-11 in Ghassemi et al., 2019, https://arxiv.org/pdf/1907.04394.pdf) that exactly describes all the constraints involved in this SR-ST problem. This long list of constraints are not re-stated in this paper, due to the amount of space that it would take in this ICLR paper.
>
> 2. Inputs and Outputs of the CAM model, and how it works: The newly added first paragraph of Section 3 now clearly describes how the CAM model works, and the inputs and outputs of the model. This description is supported by a new diagram (Fig. 1 in the revised manuscript), illustrating examples of how the CAM model is used by robots to select tasks during an operation (also summarized in our earlier response to Reviewer 1). In addition, a formal definition of the problem as MDP has been added in Section 2.3 to clearly define the state and action spaces.

---

> ### Author Response · Authors · 2020-11-24
> **2. MDP problem and state/action spaces now clearly described in revision).**
>
> We agree that these important elements were missing in the original manuscript. As stated in our prior responses, the mathematical description of the MRTA problem and the MDP over which learning occurs has now been clearly stated in the revised manuscript, as a new Section 2.

---

> ### Author Response · Authors · 2020-11-24
> **3. Adaptation of the AM method (to MRTA) for comparison now clearly described in revision.**
>
> A new short paragraph has been added as Sub-Section 5.3 in the revised manuscript to summarize how we have modified the AM implementation to apply it to the concerned MRTA class of problems, for comparison with our CAM method.

---

> ### Author Response · Authors · 2020-11-24
> **4. Contributions now explicitly stated at the end of Introduction section in the revision.**
>
> The methodological contributions of our work have now been clearly stated in Section 1.3 of the revised manuscript, and highlighted in blue font. In addition, majority of the Introduction section (Section 1) had now been modified to better articulate the context and value of the class of MRTA problems being solved here.

---

> ### Author Response · Authors · 2020-11-24
> **5. Motivation for graph representation of problem and incorporation of distance information explained.**
>
> The class of MRTA problems tackled here can be perceived as a more complex variation of the standard multi-Traveling Salesman Problem (m-TSP), and recent literature [1-3] has shown the effectiveness of representing and learning m-TSP and related combinatorial optimization problems as graphs. We refer the reader to these recent papers for more insights on the graph representation of such classes of problems. More importantly, as mentioned in our earlier responses, we have now included an entirely new Section, Section 2 (MRTA: Problem Definition and Formulations), that clearly describes the optimization problem and its ``MDP over graph" counterpart.
>
> The distance information (essentially a graph edge property) enters the decision-making of a robot through the encoder in the CAM architecture, which encodes the difference in the properties of nodes, where properties include the task location. In addition, note that the cost function over which the policy model is trained (Eq. 1) also includes the distance information.
>
>
> [1] Akash Mittal, Anuj Dhawan, Sahil Manchanda, Sourav Medya, Sayan Ranu, and Ambuj Singh. Learning heuristics overlarge graphs via deep reinforcement learning, 2019.
>
> [2] Hanjun Dai, Elias B. Khalil, Yuyu Zhang, Bistra Dilkina, and Le Song. Learning combinatorial optimization algorithms over graphs. In Advances in Neural Information Processing Systems, 2017.
>
> [3] Wouter Kool, Herke Van Hoof, and Max Welling.  Attention, learn to solve routing problems!   In7th InternationalConference on Learning Representations, ICLR 2019, 2019.

---

> ### Author Response · Authors · 2020-11-24
> **6. Comparison to standard non-RL MRTA methods explained, and mentioned potential for adding results (if needed) to further augment this explanation.**
>
> As summarized in Section 1.1 of the revised manuscript, exact solutions to larger MRTA problems are computationally prohibitive for online implementation. For example, for the studied SR-ST problem, the cost of solving the exact integer programming formulation of the problem, even with a linear cost function (thus an ILP), scales with $\mathcal{O}(n^3m^2h^2)$, where $n$, $m$, and $h$ represent the number of tasks, the number of robots, and the maximum number of tours per robot, respectively. As a result, most practical online MRTA methods, e.g., auction-based methods and bi-graph matching methods, use some sort of heuristics, and often report the optimality gap at least for smaller test cases compared to the exact ILP solutions. To put things into real-time performance perspective, consider that a related ILP formulation of the SR-ST takes 50 mins to provide the optimal solution, vs. 50 seconds taken by a standard bi-graph matching method (that involves heuristics) vs. 0.8 seconds taken by our learnt CAM policy model (all estimations are based on our numerical experiments conducted on a desktop workstation).
>
> We agree that including comparisons to state-of-the-art non-RL methods for solving MRTA would help provide better context and motivation for the use of the RL approach to learn the MRTA heuristics. Given the specific case study settings and the non-linear cost function used here, over the past week, we have had to make some modifications to available standard implementations of the corresponding exact integer programming approach (offline method) and a bi-graph matching based approach (online method). These comparative techniques are being run as we speak, and if the results are completed and analyzed by tomorrow, we will include them in a further modified version of the manuscript. If they take more time to generate, we will have to include them in the final version of the paper, i.e., if the paper gets accepted. However, we believe the paper can stand on its own without needing to include these non-RL results.

---

> ### Author Response · Authors · 2020-11-24
> **Typos fixed.**
>
> Suggested change has been implemented.

---

> ### Author Response · Authors · 2020-11-24
> **Complexity of case study at par with related problems (e.g., mTSP and VRP) used in other published work in the domain of RL over graphs**
>
> From an engineering stand-point, the MRTA problem studied here is a simplification of multi-robot delivery and multi-vehicle-routing type applications. The simplifications are mostly in terms of simplified system dynamics and environment modeling (treated here as a discrete event simulation) which do not impact the mathematical complexity of the underlying MDP (in terms of size of state/action spaces), as decisions only occur at nodes of the graph (task locations and depots). Rather, characteristics such as range/payload constraints, task deadlines and allowance of multiple tour per robot make our simulated problem setting a bit more more complex compared to (related) toy mTSP and VRP problems studied in other existing work on using RL to solve combinatorial optimization over graphs. Moreover the size of the problem that we study here, i.e., 20 robot / 200 tasks lies on the medium-high end of the sizes of MRTA problems tackled in the multi-robotics domain. It is however important to note that if a simulation environment with a smaller reality gap compared to real-world application is used, the cost of training would become significantly higher. While our results show that the presented CAM method provides faster convergence on the simplified MRTA problem in terms of number of epochs (compared to AM), which would be beneficial in a more realistic learning setup, further work is certainly needed to alleviate these higher computational costs in real-world problems.

---

### Official Review · AnonReviewer1 · 2020-10-29
**Well written paper discussing multi-robot task allocation expanding existing robot task allocation method into the multi-robot case, however description of th test case experimenting the method quite limited.**

**Rating:** 7
**Confidence:** 4

**Review:**

I recommen the paper to be accepted. It is well written paper addressing an imporant use case, allocating multi-robot tasks, e.g. UAVs in the case of flood disaster. Introduction gives credit to previous work and motivation for the research. Paper discusses the development of a method called Covariant Attention-based model (CAM) which expands the work of Kool et al (2019) into multi-robot tasks.  The theoretical background of the method is sufficiently explained and the novelty of the method is clear. Obtained results are compared against a state-of-the-art method and shown to outperform it.

The deficiency of the paper is the explanation of the experiments. It would interesting to understand how the allocation process is learned to be done, namely how would the learned process be applicable for a real scenario. Now it was only said that the area is one square kilometer and the task times vary, but more details of the simulations would give more insight ito the real usefullness of the method.

Minor corrections:
Section 4.2 1 x 1 sq.km => I guess this should be 1 sq. km or 1x1 km?
Algorithm 1:  N epochs -> N epoch (r3)

---

> ### Author Response · Authors · 2020-11-24
> **Clarifying how the learned model operates and the experimental setup used for testing.**
>
> We agree that the original manuscript left opportunities for better describing how the CAM model works in operation.  Hence, to clarify how the proposed policy architecture is used by each robot to perform task allocation, a new illustration has been added to the revised manuscript as Fig. 1, and further described at the start of Section 3.  As described therein, the allocation process performed by the CAM model can be summarized as follows:  “The CAM model for task allocation is called by for each robot right when it reaches its current destination (task location or depot), in order to decide its next task or destination.  Here, the inputs to the CAM model includes1)the task graph information (i.e., the location of each task and its associated time deadline),2)the current mission time,3)the state of robot-r, and4)the states of robot-r’s peers.  The CAM model then generates the probability of selecting each task as its output.  A greedy policy to choose the task with the highest probability is used here, which thus provides the next destination to be visited by that robot.  It should be noted that the probability values for completed tasks and missed tasks(i.e., missed deadline) are set at 0.”In addition,  Section 5.2 (Design of Experiments & Learning Procedures) has been modified and extended to better describe the case study simulations, and point out that the CAM method can generalize beyond this multi-UAV flood-response application (which is mainly used for motivation).

---

> ### Author Response · Authors · 2020-11-24
> **Typos fixed.**
>
> Thank you for pointing out the typos.  These typos have been fixed.

---

### Author Response · Authors · 2020-11-24
**Revised manuscript uploaded, and initial responses to review comments added in the discussion system.**

The revised manuscript has been uploaded. We appreciate the thoroughness of the review process and would like to thank you for giving us the opportunity to address the concerns raised by the reviewers. Furthermore, we would like to express our gratitude to the reviewers for their time and guidance. We have taken great care to precisely and clearly address their comments through the online response system, and would look forward to any further feedback that the reviewers have on the revisions made to our manuscript. Please note the revised manuscript reflects the changes/additions, highlighted using blue font.
In addition, note that the changes made in the revision are w.r.t. the presentation and discussion of the motivation for the work, its primary contributions, and key findings. Here, our goal was to answer the questions, need for clarity, and in few cases potential mis-intetpretations, raised in the review comments. Please note that no methodological changes or additions of results have been performed, thus carefully preserving the sanctity of the work presented in the original manuscript.

---

### Decision · Program_Chairs · 2021-01-07
**Final Decision**

**Decision:**

Reject

**Comment:**

This paper presents a GNN architecture for policies that solve multi-robot task allocation problems. The proposed architecture extends Koul et al (2019) by adding payload constraints and task deadlines. The paper looks at routing problems of medium-to-large size, e.g. 20 robots and 200  tasks. The reviewers are happy that most of their concerns were addressed by they are still concerned that the experimental validation is focusing too much on the multi-TSP or Vehicle Routing Problems, and request more extensive experimental validation on similar optimization problems as in Nunes et al (2017). I tend to agree. The proposed method has a lot of merit and just needs one more iteration of improvements to incorporate further experiments, before it becomes ready for publication.